# Matching Feature Separation Network for Domain Adaptation in Entity Matching

## ABSTRACT

Entity matching (EM) determines whether two records from different data sources refer to the same real-world entity. Currently, deep learning (DL) based EM methods have achieved state-of-the-art (SOTA) results. However, applying DL-based EM methods often costs a lot of human efforts to label the data. To address this challenge, we propose a new domain adaptation (DA) framework for EM called **M**atching **F**eature **S**eparation **N**etwork (MFSN). We implement DA by separating private and common matching features. Briefly, MFSN first uses three encoders to explicitly model the private and common matching features in both the source and target domains. Then, it transfers the knowledge learned from the source common matching features to the target domain. We also propose an enhanced variant called **F**eature **R**epresentation and **S**eparation **E**nhanced MFSN (MFSN-FRSE). Compared with MFSN, it has superior feature representation and separation capabilities. We evaluate the effectiveness of MFSN and MFSN-FRSE on twelve transferring EM tasks. The results show that our framework is approximately 7% higher in F1 score on average than the previous SOTA methods. Then, we verify the effectiveness of each module in MFSN and MFSN-FRSE by ablation study. Finally, we explore the optimal strategy of each module in MFSN and MFSN-FRSE through detailed tests.

## CCS CONCEPTS

• Insert CCS text here • Insert CCS text here • Insert CCS text here

## KEYWORDS

Entity matching, Deep neural network, Domain adaptation, Matching feature separation network, Data integration

**ACM Reference format:**

FirstName Surname, FirstName Surname and FirstName Surname. 2023. Matching Feature Separation Network for Domain Adaptation in Entity Matching. In *Proceedings of ACM Woodstock conference (WOODSTOCK'18). ACM, New York, NY, USA, 2 pages.*

*WOODSTOCK'18. Iune. 2018. El Paso. Texas USA*

https://doi.org/10.1145/1234567890

## 1 Introduction

Entity Matching (EM) aims to determine whether two records from different data sources refer to the same real-world entity [1]. It has been a core problem in data integration. Early works focus on using manually defined matching rules [2] or machine learning [3] for EM. In recent years, with the development of deep learning (DL), more and more researchers have used DL to solve the EM problem. Such as record distributed representation-based method (DeepER [4]), RNN and Attention based method (DeepMatcher [1]), and pre-trained language models (pre-trained LMs) based method (Ditto [5]). Among these methods, DL-based methods often achieve the best results. However, it is well known that DL-based methods are data hungry. Therefore, DL-based EM methods often cost a large amount of human effort to label the data. This manual labeling work is both expensive and time-consuming, making it impractical for many practical applications.

To improve the above issue, some works attempt to reduce dependence on labeled data by using pre-trained LMs [5, 6]. Pre-trained LMs (such as BERT [7], DistilBERT [8], RoBERTa [9]) are pre-trained on a large corpus and can provide deeper language understanding than traditional word embeddings. This excellent language understanding ability can help models generalize better to new tasks. Thus, introducing pre-trained LMs to solve EM tasks can reduce the dependence on labeled record pairs. However, even the most famous work Ditto [5] still needs at least several thousand labeled pairs to achieve satisfactory results. Later, some works start to consider using domain adaptation (DA) to reduce the cost of data labeling [10]. DA, a case of transfer learning, uses models trained on labeled data from related domains (usually called the source domain) to solve new tasks without labeled data (usually called the target domain). The main challenge of DA is the domain shift problem, as shown in Figure 1(a). Since the source and target datasets may come from different domains, they don't follow the same distribution. Therefore, the EM model trained on the source dataset doesn't correctly work on the target dataset. Recently, some DA-based EM methods have been proposed [10, 11]. These methods only directly learn the common features in source and target domains by applying special constraints on the feature mapping function. However, these methods ignore explicit modeling of domain-private features. Domain-private features are only meaningful in a specific domain, such as technical terms or customary usage in a domain. As shown in Figure 1(b), if these features are not differentiated and processed, the matching decision stage may be disturbed by the discrepancy between the source and target

domains. Therefore, the main challenge of DA in EM is that we need to explicitly model the domain-private features so that the final common features contain as few domain-private properties as possible.

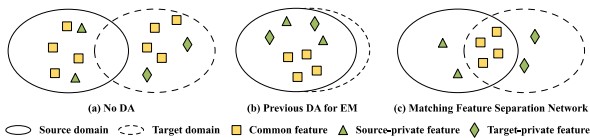

Figure 1: Domain adaptation (DA) for entity matching (EM).

To address this challenge, we propose a new DA framework for EM called Matching Feature Separation Network (MFSN). As shown in Figure 1(c), MFSN can learn better common matching features by explicitly modeling the private matching features of each domain. We conduct various experiments on twelve transferring EM tasks, and the results indicate the effectiveness of our proposed model. The main contributions of this paper are as follows:

1. We propose a framework called **M**atching **F**eature **S**eparation **N**etwork (MFSN). MFSN explicitly models the private and common matching features in the source and target domains by three encoders and transfers only the knowledge in common matching features.

2. We propose an enhanced variant of MFSN called **F**eature **R**epresentation and **S**eparation **E**nhanced MFSN (MFSN-FRSE). Compared with MFSN, MFSN-FRSE has better feature representation and separation capabilities.

3. We evaluate the effectiveness of MFSN and MFSN-FRSE on twelve transferring EM tasks (six similar domain tasks and six different domain tasks). Our experimental results show that the MFSN-FRSE is approximately 7% higher in F1 score on average than the previous SOTA methods. Then, we verify the effectiveness of each module in MFSN and MFSN-FRSE by ablation study. Finally, we explore the optimal strategy of key modules in MFSN and MFSN-FRSE by detail tests.

The rest of this paper is organized as follows. Section 2 introduces the related work about EM, transfer learning, and transferring EM. Section 3 first defines the EM task and DA in EM, and then proposes a framework for DA in EM. Section 4 reports a series of comparative experiments. Section 6 concludes the paper.

## 2 Related Work

Early works on EM are devoted to designing various matching rules [2]. However, these methods lack universality, as no matching rules are suitable for all datasets. In recent years, DL-based EM methods have been extensively studied and have achieved the SOTA results. DeepMatcher [1] proposes a DL-based EM framework, which includes four solutions with varying representational power: SIF, RNN, Attention, and Hy-

brid. Ditto [5] applies pre-trained LMs to EM tasks, which achieves the SOTA results and reduces the number of training data. DL-based EM methods can automatically generate more expressive features and satisfy the end-to-end needs of real-world applications. However, DL-based methods still need a large amount of training data to achieve satisfying results.

Transfer learning (TL) refers to transferring knowledge learned in an old domain to a new domain by utilizing similarities between data, tasks, or models [12]. In the EM literature, only a few studies have focused on TL. Kirielle et al. [18] propose an instance-based method, called TransER. This method first selects source instances with similar features and neighborhoods to the target domain instances. Then, it uses selected source instances to train a classifier that can work on the target domain. However, this method uses attribute-based similarities to generate features, so it is only applicable to the structured dataset and can't be applied to the unstructured dataset. Tu et al. [10] propose a DA framework for EM, called DADER. This framework systematically explores the design space and compares different choices of DA for ER. Some methods in DADER achieve the current SOTA results. However, these methods in DADER directly learn common features through statistics metrics or adversarial training. By disregarding the explicit modeling of domain-private features, it can't be guaranteed that the final common features contain as few domain-private properties as possible. Sun et al. [19] propose a DSN-based method called VAER-DSN. VAER-DSN uses gated recurrent units (GRU) and variational auto-encoders (VAE) as the basic components to learn the private and common features. However, the pre-trained LMs are not considered as the underlying models. As mentioned earlier, they have general natural language comprehension capabilities, therefore they can be a good starting point to help the models quickly adapt to a new task. Thus, in the next section, we will utilize the pre-trained LMs to solve the DA in EM.

## 3 Matching Feature Separation Network for Domain Adaptation in Entity Matching

### 3.1 Task Definitions

Entity matching (EM) aims to determine whether two records from different data sources refer to the same entity [1]. Let $D_1$ and $D_2$ be two collections of entity records with multiple attributes. Each record $r_1 \in D_1$ (or $r_2 \in D_2$) is a set of key-value pairs $\{attr_i, val_i\}_{1 \leq i \leq k}$, where $attr_i$ and $val_i$ denote the $i$-th attribute name and attribute value respectively. EM aims to determine whether $r_1$ and $r_2$ refer to the same real-world entity or not. A typical EM pipeline consists of two steps: blocking and matching. The blocking step generates a candidate set $Cnd \subset D_1 \times D_2$ with a high recall by removing unnecessary comparisons. The subsequent matching step only needs to determine whether the candidate pair $(r_1, r_2) \in Cnd$ match or not. The DL-based EM method first defines an EM model $\mathcal{M}$, which takes the candidate pair $(r_1, r_2)$ as input and outputs a prediction $\hat{y}$.

$$\hat{y} = \mathcal{M}(r_1, r_2) \tag{1}$$

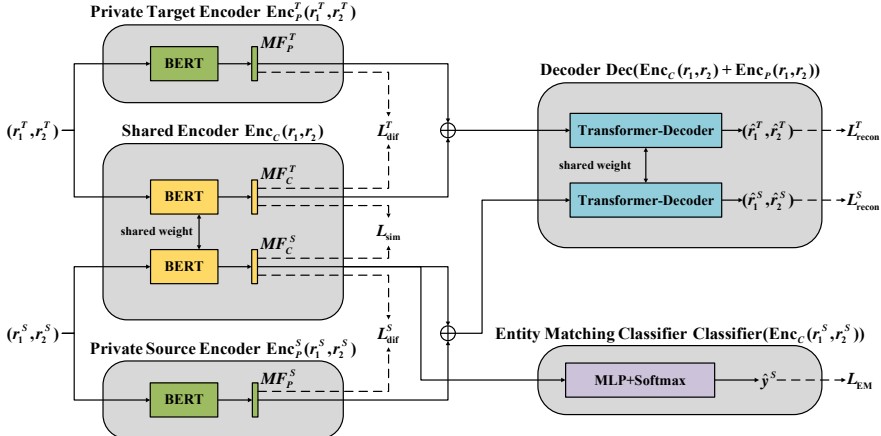

**Figure 2: The architecture of Matching Feature Separation Network (MFSN).**

Subsequently, a training set $(D, Y) = \{(r_1, r_2, y)\}$ is defined, where $D \subset D_1 \times D_2$ is the set of record pairs and $Y$ is the set of labels. Finally, the $\mathcal{M}$ is trained on the $(D, Y)$, and its parameters are updated such that it can correctly determine whether the input record pairs match or not.

Next, we define domain adaptation (DA) in EM. Given a labeled source dataset $(D^S, Y^S) = \{(r_1^S, r_2^S, y^S)\}$ and an unlabeled target dataset $D^T = \{(r_1^T, r_2^T)\}$, the goal is to train an EM model with labeled source data and unlabeled target data, so that it does not only work well on $D^S$, but also is able to make correct predictions on $D^T$.

## 3.2 Framework Overview

We propose a framework for DA in EM called **M**atching **Fe**ature **S**eparation **N**etwork (MFSN). As shown in Figure 2, MFSN consists of five modules: private target encoder $\text{Enc}_P^T$, private source encoder $\text{Enc}_P^S$, shared encoder $\text{Enc}_C$, decoder Dec, and EM classifier Classifier. The two private encoders are used to learn the private matching features in the source and target domains respectively, while the shared encoder is used to learn the common matching features in both domains. The decoder takes the sum of private and common matching features as input to reconstruct the input candidate pairs. The EM classifier makes matching decisions with the common matching features. For simplicity, we use $\text{Enc}_P$ to denote the private encoder, disregarding its domain.

## 3.3 Encoder for Matching Feature Separation

Encoder Enc aims to learn the matching features $MF$ from the candidate pairs $(r_1, r_2)$, which can be used for subsequent matching decisions.

$$MF = \text{Enc}(r_1, r_2) \tag{2}$$

The effectiveness of DA largely depends on the quality of $MF$, so the choice of encoder architecture is very important. There

are many choices of encoder architectures: RNN, Attention, VAE, and pre-trained LMs. We choose BERT [7], a Transformer-Encoder-based pre-trained LM, as the encoder for MFSN. Firstly, the pre-trained LMs (such as BERT) have excellent language compression capability, which can provide strong support for DA.

*3.3.1 Encoder Sketch.* The procedure to obtain $MF$ by using BERT [7] as an encoder is as follows:

First of all, given a candidate pair $(r_1, r_2)$ and each record $r = \{attr_i, val_i\}_{1 \le i \le k}$. The candidate pair $(r_1, r_2)$ is converted into a candidate pair sequence $S(r_1, r_2)$ using Equation (3) and Equation (4):

$$S(r_1, r_2) = [\text{CLS}]S(r_1)[\text{SEP}]S(r_2)[\text{SEP}] \tag{3}$$

$$S(r) = [\text{ATT}]attr_1[\text{VAL}]val_1 \dots [\text{ATT}]attr_k[\text{VAL}]val_k \tag{4}$$

[ATT] and [VAL] are used to indicate the beginning of the attribute name and attribute value, [SEP] is used to separate the two records, and [CLS] is used to encode the whole sequence.

Then, $S(r_1, r_2)$ is input to BERT for encoding, and the hidden representation of [CLS] is obtained as $MF$.

As shown in Figure 2, MFSN defines three encoders. The two private encoders are used to learn private matching features of the source and target domains respectively, as shown in Equation (5). The shared encoder is used to learn common (i.e., domain-invariant) matching features of each domain, as shown in Equation (6).

$$MF_P^S, MF_P^T = \text{Enc}_P^S(r_1^S, r_2^S), \text{Enc}_P^T(r_1^T, r_2^T) \tag{5}$$

$$MF_C^S, MF_C^T = \text{Enc}_C(r_1^S, r_2^S), \text{Enc}_C(r_1^T, r_2^T) \tag{6}$$

$MF_P^S \in \mathbb{R}^d$ ($MF_P^T \in \mathbb{R}^d$) denotes the private matching features of the source (target) domain, and the $MF_C^S \in \mathbb{R}^d$ ($MF_C^T \in \mathbb{R}^d$) denotes the common matching features of the source (target) domain. For simplicity, we use $MF_C$ ($MF_P$) to denote the common (private) matching feature, disregarding its domain.

*3.3.2 Similarity Loss.* The similarity loss $L_{\text{sim}}$ can effectively measure the discrepancy between the feature distributions of $MF_C^S$ and $MF_C^T$.

$$L_{\text{sim}} = \text{Distribution\_Discrepancy}(MF_C^S, MF_C^T) \qquad (7)$$

By continuously minimizing $L_{\text{sim}}$, the distributions of $MF_C^S$ and $MF_C^T$ are becoming more and more similar. In this paper, we propose three methods to implement $L_{\text{sim}}$, denoted as MMD loss [13], CORAL loss [14], and GRL loss [15].

*MMD Loss.* The MMD loss is based on Maximum Mean Discrepancy (MMD) [13]. Briefly, MMD maps the distributions into the reproducing kernel Hilbert space (RKHS) by using a kernel function. Then MMD takes the distance between the mean embeddings of the two distributions in the RKHS as the discrepancy between them:

$$L_{\text{sim}}^{\text{MMD}} = \sup_{\|\varphi\|_H \leq 1} \Big\| \mathbb{E}_{MF_C^S \sim \text{Enc}_C(D^S)}[\varphi(MF_C^S)] -$$
$$\mathbb{E}_{MF_C^T \sim \text{Enc}_C(D^T)}[\varphi(MF_C^T)] \Big\|_H^2 \qquad (8)$$

$\varphi(\cdot)$ represents a kernel function that maps $MF_C^S$ ( $MF_C^T$) to an RKHS, and $\|\varphi\|_H \leq 1$ defines a set of functions in the unit ball of RKHS. $L_{\text{sim}}^{\text{MMD}} = 0$ if and only if the distributions of $MF_C^S$ and $MF_C^T$ are the same.

*CORAL Loss.* The CORAL loss is based on CORrelation Alignment [14], which is a particular case of $k$-order where $k = 2$. It measures the discrepancy between two distributions by computing the difference between their covariance matrices (second-order statistics). The CORAL loss is defined as Equation (9).

$$L_{\text{sim}}^{\text{Coral}} = \frac{1}{4d^2} \big\| \text{cov}(\mathbf{MF}_C^S) - \text{cov}(\mathbf{MF}_C^T) \big\|_F^2 \qquad (9)$$

$\mathbf{MF}_C^S(\mathbf{MF}_C^T) \in \mathbb{R}^{n \times d}$ is the common matching feature matrix of the source (target) domain, where $d$ is the dimensionality of $MF_C$ and $n$ is the number of source (target) samples. The function $\text{cov}(\cdot)$ is used to compute the covariance matrix for a given matching features matrix. $\|\cdot\|_F^2$ denotes the squared matrix Frobenius norm.

*GRL Loss.* The GRL loss is based on adversarial training [15]. We first introduce a Discriminator to determine whether the $MF_C$ are from the source or target domain, as shown in Equation (10). Then, the Discriminator and $\text{Enc}_C$ are trained in an adversarial manner: the $\text{Enc}_C$ aims to maximize the domain classification error, while the Discriminator aims to minimize it. The loss function is shown in Equation (11), and the adversarial training is implemented by adding a gradient reversal layer (GRL) [15] between the $\text{Enc}_C$ and Discriminator:

$$\hat{d} = \text{Discriminator}(MF_C) \qquad (10)$$

$$L_{\text{sim}}^{\text{GRL}} = \sum_{i=0}^{N_S + N_T} \big\{ d_i \log \hat{d}_i + (1 - d_i) \log(1 - \hat{d}_i) \big\} \qquad (11)$$

$N_S$ and $N_T$ denote numbers of common matching features in the source and target domains, respectively, and $d_i \in \{0,1\}$ denotes the domain label for $MF_C$.

*3.3.3 Difference Loss.* The difference loss [16] can help the private and shared encoders to encode different aspects of the input. Given a source or target domain candidate pair $(r_1, r_2)$, we first use the corresponding private and shared encoder to obtain $MF_P$ and $MF_C$. The difference loss is defined as Equation (12).

$$L_{\text{dif}} = \big\| MF_P^\top MF_C \big\|_F^2 \qquad (12)$$

$\|\cdot\|_F^2$ denotes the squared matrix Frobenius norm. Minimizing $L_{\text{dif}}$ can help the shared and private encoders generate mutually orthogonal features.

## 3.4 Decoder for Reconstruction

The decoder Dec uses both common and private matching features to reconstruct the original input pairs. As shown in Figure 2, given a source or target domain candidate pair $(r_1, r_2)$, we first use the corresponding private and shared encoder to obtain $MF_P$ and $MF_C$, respectively. Then $MF_P$ and $MF_C$ are summed up and fed into the Dec for decoding:

$$(\hat{r}_1, \hat{r}_2) = \text{Dec}(MF_C + MF_P) \qquad (13)$$

We evaluate the effectiveness of the two encoders by comparing the discrepancy between $(\hat{r}_1, \hat{r}_2)$ and $(r_1, r_2)$. Finally, we use the discrepancy as the reconstruction loss to further optimize the model. The reconstruction loss is defined as:

$$L_{\text{recon}} = \text{Record\_Discrepancy}\big((\hat{r}_1, \hat{r}_2), (r_1, r_2)\big)$$
$$= \sum_{i=1}^{m} \text{CE}(\hat{t}_i, t_i) \qquad (14)$$

The function $\text{CE}(\cdot)$ is the Cross-Entropy loss function. The $t_i$ denotes the $i$-th token in candidate pairs sequence $S(r_1, r_2)$ which is obtained from $(r_1, r_2)$ by Equation (3). The $\hat{t}_i$ denotes the $i$-th token in $S(\hat{r}_1, \hat{r}_2)$, and $m$ denotes the length of $S(r_1, r_2)$.

For the choice of decoder architecture, we can't directly use the vanilla Transformer-Decoder architecture [21]. As shown in Equation (15), if the encoder only provides a single vector $MF$ for decoding, the decoder's CrossAttention layer always outputs the same value for any query vector $q$, which comes from the previous masked self-attention layer.

$$MF^\top \mathbf{W}_V = \text{CrossAttention}(q, MF^\top \mathbf{W}_K, MF^\top \mathbf{W}_V) \qquad (15)$$

$\mathbf{W}_K$ and $\mathbf{W}_V$ are key and value matrices. To avoid the above situation, we add a gate mechanism [22] to the vanilla Transformer-Decoder: Suppose $q_t$ is the $t$-th query vector, and the output $o_t$ of the CrossAttention layer is:

$$o_t = \sigma(q_t \mathbf{W}_1 + MF \mathbf{W}_2) \odot MF^\top \mathbf{W}_V \qquad (16)$$

The function $\sigma(\cdot)$ is the sigmoid activation function, $\mathbf{W}_1$ and $\mathbf{W}_2$ are two learnable parameter matrices, $\mathbf{W}_V$ is value matrix, and $\odot$ denotes Hadamard product.

## 3.5 Entity Matching Classifier and Objective Function

EM classifier can make matching decisions based on $MF_C$.

$$\hat{y} = \text{Classifier}(MF_C) \tag{17}$$

We utilize the $MF_C^S$ to train an EM classifier, while the EM loss is defined as Equation (18).

$$L_{EM} = \sum_{i=0}^{N} y_i \log \hat{y}_i + (1 - y_i) \log(1 - \hat{y}_i) \tag{18}$$

$N$ denotes the number of candidate pairs in the source domain, $y_i$ denotes the label for the $i$-th pairs, and $\hat{y}_i$ is the prediction made by EM classifier for it.

The final objective function consists of EM loss, reconstruction loss, difference loss, and similarity loss, as shown in Equation (19). Our training goal is to minimize this objective function.

$$L = L_{EM} + \alpha L_{\text{recon}} + \beta L_{\text{dif}} + \gamma L_{\text{sim}} \tag{19}$$

$\mathcal{L}_{\text{EM}}$ comes from Section 3.5, $\mathcal{L}_{\text{recon}}$ comes from Section 3.4, $\mathcal{L}_{\text{dif}}$ and $\mathcal{L}_{\text{sim}}$ comes from Section 3.3.

## 3.6 Feature Representation and Separation Enhancement

In this section, we propose an enhanced variant called **F**eature **R**epresentation and **S**eparation **E**nhanced MFSN (MFSN-FRSE). First of all, to enhance the feature representation capability, we propose an enhanced encoder to obtain the hidden representation of all tokens. Then, the difference loss is computed in a "token-by-token" manner, the similarity loss is computed with the help of DomAtt. Lastly, the decoder takes all tokens' private and common features as input to reconstruct the original input pairs. In MFSN-FRSE, we mainly improve the encoder and decoder modules, while the other modules are the same as MFSN.

*3.6.1 Enhanced Encoder.* Recall that the encoder in MFSN simply uses the hidden representation of [CLS] as $MF$, which is suitable for classification tasks. However, TL tasks often require a higher feature representation capability of the model. Therefore, to improve the feature representation capability, we propose an enhanced encoder EEnc. Specifically, it uses BERT to obtain the hidden representations of all tokens in $(r_1, r_2)$, and combine them to form a hidden representation matrix **H**:

$$\mathbf{H} = \text{EEnc}(r_1, r_2) \tag{20}$$

The hidden representation matrix $\mathbf{H} \in \mathbb{R}^{m \times d}$, $m$ is the number of tokens in candidate pairs sequence $S(r_1, r_2)$ obtained from $(r_1, r_2)$ by Equation (3), and $d$ is the dimensionality of the BERT's output. As shown in Figure 3 and Figure 4, we next use a token sequence-based method to compute the difference loss and similarity loss.

*Enhanced Similarity Loss.* The enhanced similarity loss can help the enhanced shared encoder $\text{EEnc}_C$ learn similar common features from the source and target domain. We introduce a pooling layer Pool to efficiently compute the similarity loss between two hidden representation matrices. As shown in Figure 3, we first obtain the common hidden representation

matrices $\mathbf{H}_C^S$ and $\mathbf{H}_C^T$ of $(r_1^S, r_2^S)$ and $(r_1^T, r_2^T)$ by the $\text{EEnc}_C$. Next, we use the Pool to learn a domain-aware matching feature vector from the $\mathbf{H}_C^S$ and $\mathbf{H}_C^T$, respectively:

$$h_C^S, h_C^T = \text{Pool}(\mathbf{H}_C^S), \text{Pool}(\mathbf{H}_C^T) \tag{21}$$

The similarity loss of $h_C^S$ and $h_C^T$ is then computed to ensure that their feature distributions are similar. The similarity loss between $h_C^S$ and $h_C^T$ can be computed in three ways described in Section 3.3.

However, conventional pooling strategies (such as using special tokens or averaging operations) can't accurately capture the domain information of the sequence. So, we propose a DomAtt mechanism, which is based on self-Attention [21]. As shown in Equation (22)-(23), the DomAtt takes a hidden representation matrix as the key and the value, while a learnable domain-shared vector $a$ as the query.

$$h_c^T = \text{softmax}\left(\frac{a^T \mathbf{W}^Q (\mathbf{H}_C^T \mathbf{W}^K)^\top}{\sqrt{d}}\right) \mathbf{H}_C^T \mathbf{W}^V \tag{22}$$

$$h_c^S = \text{softmax}\left(\frac{a^S \mathbf{W}^Q (\mathbf{H}_C^S \mathbf{W}^K)^\top}{\sqrt{d}}\right) \mathbf{H}_C^S \mathbf{W}^V \tag{23}$$

The learnable parameter matrices $\mathbf{W}^Q, \mathbf{W}^K, \mathbf{W}^V \in \mathbb{R}^{d \times d}$, and $d$ is the dimensionality of the input features. For the technical details of the self-attention mechanism, please refer to [21].

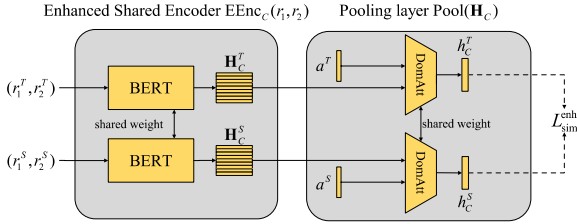

**Figure 3: The way to compute the enhanced similarity loss, in MFSN-FRSE. $a^S$ and $a^T$ are two learnable domain-shared vectors in the source and target domains, respectively.**

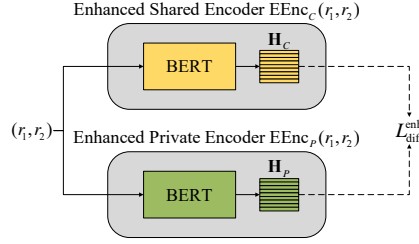

**Figure 4: The way to compute the enhanced difference loss, in MFSN-FRSE.**

*Enhanced Difference Loss.* The enhanced difference loss can help the enhanced private encoder $\text{EEnc}_P$ and enhanced

shared encoder $\text{EEnc}_C$ to encode different aspects of the input. As shown in Figure 4, Given a source or target domain candidate pair $(r_1, r_2)$, we first obtain the private hidden representation matrix $\mathbf{H}_C$ and common hidden representation matrix $\mathbf{H}_P$, respectively. Then, we adopt a token-by-token manner to compute the difference loss:

$$L_{\text{dif}}^{\text{enh}} = \sum_{i=1}^{m} \|\mathbf{H}_C[i]^\top \mathbf{H}_P[i]\|_F^2 \qquad (24)$$

$\mathbf{H}_C[i]$ ($\mathbf{H}_P[i]$) denotes the $i$-th row of the hidden representation matrix $\mathbf{H}_C$ ($\mathbf{H}_P$), which corresponds to the hidden representation of the $i$-th token in the $\text{S}(r_1, r_2)$ obtained from $(r_1, r_2)$ by Equation (3). $m$ denotes the length of $\text{S}(r_1, r_2)$.

*3.6.2 Enhanced Decoder.* Decoder Dec takes $\mathbf{H}_c$ and $\mathbf{H}_P$ as inputs to reconstruct the original input candidate pairs. As mentioned before, it can help the $\text{EEnc}_C$ and $\text{EEnc}_P$ to learn more effective features. Specifically, given a source or target domain candidate pair $(r_1, r_2)$, we first use the corresponding $\text{EEnc}_C$ and $\text{EEnc}_P$ to obtain the $\mathbf{H}_P$ and $\mathbf{H}_c$. Then the $\mathbf{H}_P$ and $\mathbf{H}_c$ are summed up and fed into the Dec for decoding:

$$(\hat{r}_1, \hat{r}_2) = \text{Dec}(\mathbf{H}_P + \mathbf{H}_C) \qquad (25)$$

The reconstruction loss can be computed using Equation (14) in Section 3.4. For the architecture of Dec, we use a single layer of Transformer-Decoder.

# 4 Experimental Evaluation

## 4.1 Experiment Setup

*4.1.1 Datasets.* Table 1 displays the statistical information of all datasets used in the experiment. The first nine datasets are obtained from DeepMatcher [1] and cover a wide range of domains such as products, citations, and restaurants. The latter four datasets are obtained from the WDC product dataset [23]. This dataset collects data from multiple ecommerce sites and categorizes them into four categories: computers, watches, shoes, and cameras. Each category has 1,100 labeled candidate pairs. Finally, we denote a DA task by $D^S \rightarrow D^T$, where $D^S$ is the source dataset and $D^T$ is the target dataset.

**Table 1: Datasets used in our experiments. #Pairs, #Matches, and #Attrs represent the numbers of entity pairs, matching pairs, and attributes, respectively.**

| Dataset | Domain | #Pairs | #Matches | #Attrs |
|---|---|---|---|---|
| Walmart-Amazon (WA) | Product | 10,242 | 962 | 5 |
| Abt-Buy (AB) | Product | 9,575 | 1,028 | 3 |
| DBLP-Scholar (DS) | Citation | 28,707 | 5,347 | 4 |
| DBLP-ACM (DA) | Citation | 12,363 | 2,220 | 4 |
| Fodors-Zagats (FZ) | Restaurant | 946 | 110 | 6 |
| Zomato-Yelp (DZY) | Restaurant | 894 | 214 | 3 |
| iTunes-Amazon (IA) | Music | 532 | 132 | 8 |
| RottenTomatoes-IMDB (RI) | Movies | 600 | 190 | 3 |
| Books2 (B2) | Books | 394 | 92 | 9 |
| WDC-Computers (CO) | Product | 1,100 | 300 | 2 |
| WDC-Cameras (CA) | Product | 1,100 | 300 | 2 |
| WDC-Watches(WT) | Product | 1,100 | 300 | 2 |
| WDC-Shoes (SH) | Product | 1,100 | 300 | 2 |

*4.1.2 Baselines.* To demonstrate the effectiveness of our model, we set following baselines:

*NoDA.* NoDA uses the pre-trained LMs as an encoder to learn matching features from the input candidate pair. Then, it uses a classifier to make matching decisions based on the learned features. Notice that NoDA doesn't use any DA.

*VAER-DSN* [19]. VAER-DSN is based on the DSN model. VAER-DSN uses GRU and VAE as the basic components to learn the private and common features. The classifier in VAER-DSN can make matching decisions by using the common features. The decoder can reconstruct the learned features back to the encoder input.

*DADER* [10]. DADER is a famous framework of DA for EM, and describes six representative methods: MMD, K-order, GRL, InvGAN, InvGAN+KD, and ED. The experimental results show that: 1) the methods based on pre-trained LMs often achieve the best results. 2) the results of ED are even worse than NoDA in most cases. Therefore, we use MMD, K-order, GRL, InvGAN, and InvGAN+KD as the baselines in our experiments, and all five methods adopt the pre-trained LMs.

In the subsequent experiments, we use MFSN-basic to represent the basic model introduced in Section 3.2-3.5. MFSN-basic-MMD, MFSN-basic-CORAL, and MFSN-basic-GRL represent MFSN-basic with three different similarity losses: MMD loss, CORAL loss, and GRL loss, respectively. Similarly, MFSN-FRSE is used to represent the enhanced variant introduced in Section3.6. MFSN-FRSE-MMD, MFSN-FRSE-CORAL, and MFSN-FRSE-GRL represent MFSN-FRSE with three different similarity losses, respectively.

*4.1.3 Evaluation Metric and Experiment Settings.* Following most EM works [1, 5, 10], we use precision, recall, and F1 as the evaluation metrics. Specifically, precision = |TP| / (|TP| + |FP|), recall = |TP| / (|TP| + |FN|), and F1 = 2 · precision · recall / (precision + recall). TP denotes true positives, FP denotes false positives, FN denotes false negatives, and | · | denotes the cardinal number of a set.

All experiments are implemented using Python. In all methods, the batch size is set to 32. The pre-trained LMs are uniformly using "distilbert-base-uncased" [8]. We used a server with NVIDIA GeForce RTX 3090 GPU for the experiments All experiments are repeated three times, and the average results are reported.

## 4.2 Main Results

The effect of DA may depend on the discrepancy levels between the source and target datasets, so we classify the tasks into two categories: similar domain tasks (selecting source and target domain datasets from the same domain, e.g., FZ → DZY) and different domain tasks (selecting source and target datasets from different domains, e.g., B2 → FZ). The experimental results are shown in Table 2.

From the overall perspective, the best method is MFSN-FRSE-MMD, followed by MFSN-FRSE-GRL. They significantly outperform NoDA in both task settings, which shows the effectiveness of our proposed method. Among all methods, VAER-DSN usually achieves the worst results. The main reason is

**Table 2: F1 score on similar and different domain tasks. Bold, single underline, and double underline indicate the best, second, and third values, respectively. "Avg-sim" represents the average for tasks in six similar domains. "Avg-dif" represents the average for tasks in six different domains. "Avg-all" represents the overall average across all 12 tasks.**

| | | Similar domain tasks | | | | | | | Difference domain tasks | | | | | | | Avg-all |
|---|---|---|---|---|---|---|---|---|---|---|---|---|---|---|---|---|
| | | FZ↓DZY | DZY↓FZ | SH↓CA | CA↓CO | CO↓WA | WA↓SH | Avg-sim | B2↓FZ | B2↓DZY | RI↓WA1 | RI↓AB | IA↓DA | IA↓DS | Avg-dif | |
| NoDA | | 6.27 | 82.87 | 50.75 | 69.13 | 72.85 | 65.77 | 57.84 | 49.51 | 29.55 | 18.28 | 19.34 | 74.42 | 56.15 | 41.21 | 49.52 |
| VAER-DSN | | 32.59 | 15.89 | 37.31 | 44.55 | 43.01 | 46.14 | 35.58 | 26.94 | 40.47 | 19.24 | 20.50 | 52.57 | 42.71 | 33.74 | 35.16 |
| DADER | MMD | 53.19 | 66.98 | 64.03 | 70.08 | 74.76 | 68.77 | 66.30 | 24.51 | 13.31 | 18.85 | 22.61 | 91.52 | 85.22 | 42.67 | 54.49 |
| | CORAL | 75.37 | 75.91 | 64.51 | 69.30 | 68.44 | 70.67 | 70.70 | 74.27 | 54.23 | 19.81 | 19.72 | 79.33 | 58.41 | 50.96 | 60.83 |
| | GRL | 23.98 | 83.16 | 59.34 | 70.45 | 69.13 | 63.43 | 61.58 | 62.57 | 46.18 | **34.03** | **28.46** | 85.64 | 70.24 | 54.52 | 58.05 |
| | InvGAN | 28.68 | 91.10 | 57.73 | 68.60 | 67.86 | 67.10 | 63.51 | 63.25 | 44.16 | 23.83 | 21.62 | 85.97 | 69.95 | 51.46 | 57.49 |
| | InvGAN+KD | 19.18 | 90.74 | 64.61 | 68.51 | 75.01 | 71.91 | 64.99 | 62.70 | 36.49 | 25.56 | 23.27 | 87.35 | 71.75 | 51.19 | 58.09 |
| MFSN-basic | MMD | 46.74 | 80.91 | 62.93 | 72.43 | 74.67 | 70.08 | 67.96 | 69.12 | 42.76 | 22.06 | 25.32 | 91.52 | **86.63** | 56.24 | 62.10 |
| | CORAL | 68.47 | 53.20 | 64.18 | 70.30 | 72.25 | 68.49 | 66.15 | 52.65 | 51.04 | 20.11 | 22.29 | 89.05 | 78.47 | 52.27 | 59.21 |
| | GRL | 29.06 | 78.27 | 62.30 | 72.32 | 72.01 | 73.05 | 64.58 | 84.45 | 37.47 | 29.20 | 25.56 | **91.98** | 76.20 | 57.48 | 60.99 |
| MFSN-FRSE | MMD | 58.58 | **91.67** | 67.24 | 70.16 | 76.44 | 72.56 | **72.78** | **90.00** | **60.40** | 25.01 | 25.95 | 90.44 | 85.86 | **62.94** | **67.86** |
| | CORAL | **86.12** | 58.54 | 66.05 | 68.50 | 74.09 | 67.60 | 70.15 | 51.94 | 56.70 | 28.08 | 25.26 | 87.91 | 78.40 | 54.72 | 62.43 |
| | GRL | 50.67 | 87.54 | **68.04** | **72.95** | **78.03** | **73.61** | 71.81 | 87.72 | 50.84 | 30.95 | 25.93 | 91.10 | 82.80 | 61.56 | 66.68 |

that VAER-DSN doesn't use the Transformer-Encoder-based pre-trained LMs as its underlying model. As mentioned before, these pre-trained LMs not only have powerful feature representation capabilities but also have general comprehension capabilities. Therefore, for DA in EM, it is necessary to use Transformer-Encoder-based pre-trained LMs as the underlying model.

Compared with DADER, MFSN-FRSE-MMD have higher average F1 scores in both similar and different domain tasks. Specifically, in similar domain tasks, MFSN-FRSE-MMD shows an average improvement of 2.08%. In different domain tasks, the improvement is even more significant at 8.42%. This indicates that MFSN-basic and MFSN-FRSE have more advantages in different domain tasks. Compared with similar domain tasks, the different domain tasks may have more domain-private features. Methods in DADER don't explicitly model or process these domain-private features. As a result, the learned domain-invariant features are not clean (may contain some domain-private features), which may affect the models' performances. MFSN-basic and MFSN-FRSE can separate the private features from the common features. Thus, our proposed models can achieve better DA performances.

We can find that the performance of MFSN-FRSE is generally better than MFSN-basic. In some cases, MFSN is even worse than NoDA. The possible reason is that the encoder in MFSN-basic has limited feature representation capability. This assumption can be confirmed by subsequent ablation studies. At last, the average F1 score of MMD loss is better than that of both CORAL loss and GRL loss. Therefore, we chose the MMD loss as the default similarity loss for subsequent experiments.

## 4.3 Visualization Analysis of Transferring Effect

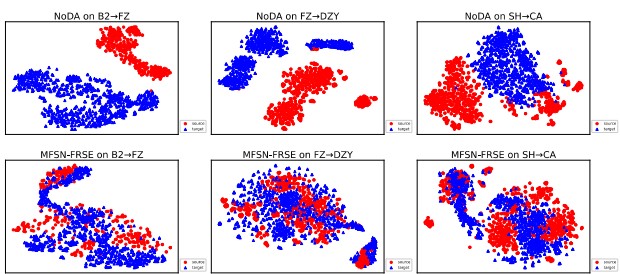

**Figure 5: DA Visualization of MFSN-FRSE. Distributions of source (red) and target (blue) are much closer in MFSN-FRSE than NoDA.**

To further analyze the effect of DA, we use t-SNE [25] to map the matching features learned from the source and target datasets into a two-dimensional space. Due to the limitation of space, we only show three representative cases. The top of Figure 5 shows the matching feature distribution obtained by NoDA, and the bottom shows the matching feature distribution obtained by MFSN-FRSE. Compared with NoDA, MFSN-FRSE generates more similar feature distributions for the source and target domain, which help the entity matching classifier make correct predictions on the target dataset.

## 4.4 Ablation Study

Next, we analyze the effectiveness of difference loss, similarity loss, and decoder by ablation study. The results are shown in Table 3-4.

**Table 3. Ablation test for MFSN-basic. Complete represents the complete model. W/o decoder, w/o difference, and w/o similarity represent removing the decoder, difference loss, and similarity loss, respectively. Bold indicates the best value.**

|  | complete | w/o decoder | w/o difference | w/o similarity |
|---|---|---|---|---|
| B2-FZ | 69.12 | 67.69 | 65.37 | **72.07** |
| B2-DZY | 42.76 | **53.10** | 34.08 | 45.29 |
| RI-WA1 | 22.06 | 18.24 | **23.93** | 23.06 |
| RI-AB | **25.32** | 24.41 | 22.57 | 23.78 |
| IA-DA | 91.52 | 92.32 | **92.81** | 73.08 |
| IA-DS | 86.63 | **87.16** | 87.05 | 66.19 |
| FZ-DZY | 47.74 | **55.88** | 54.20 | 52.43 |
| DZY-FZ | 80.91 | 80.99 | **86.33** | 71.78 |
| SH-CA | 62.93 | 63.47 | **66.85** | 63.96 |
| CA-CO | **72.43** | 70.88 | 69.92 | 70.33 |
| CO-WA | 74.67 | 72.31 | 72.76 | **74.70** |
| WA-SH | 70.08 | 67.77 | **70.84** | 70.25 |
| average | 62.18 | **62.85** | 62.23 | 58.91 |

**Table 4. Ablation test for MFSN-FRSE. Complete represents the complete model. W/o decoder, w/o difference, and w/o similarity represent removing the decoder, difference loss, and similarity loss, respectively. Bold indicates the best value.**

|  | complete | w/o decoder | w/o difference | w/o similarity |
|---|---|---|---|---|
| B2-FZ | **90.00** | 79.90 | 84.24 | 45.19 |
| B2-DZY | **60.40** | 50.11 | 37.94 | 56.84 |
| RI-WA1 | 25.01 | 23.23 | **28.08** | 19.57 |
| RI-AB | 25.95 | 25.31 | **26.22** | 21.28 |
| IA-DA | **90.44** | 90.34 | 89.53 | 63.64 |
| IA-DS | **84.86** | 84.83 | 84.02 | 57.74 |
| FZ-DZY | **58.58** | 57.02 | 56.32 | 2.05 |
| DZY-FZ | **91.67** | 90.24 | 88.35 | 84.52 |
| SH-CA | **67.24** | 64.30 | 66.71 | 57.04 |
| CA-CO | 70.16 | **73.22** | 70.19 | 70.22 |
| CO-WA | 76.44 | 73.31 | 73.29 | **77.71** |
| WA-SH | **72.56** | 70.33 | 70.90 | 70.17 |
| average | **67.78** | 65.18 | 64.65 | 52.16 |

We can see that if the decoder is removed, the MFSN-FRSE's performance will decrease. This indicates that the decoder can help the model to learn more effective features. However, if the decoder is removed, the MFSN-basic's performance will increase. The main reason is that the encoders in MFSN-basic have limited representation capability. These encoders only use the hidden representation of [CLS] as the matching feature, which can't simultaneously contain the information for matching decisions and reconstruction tasks.

In most tasks, if the difference loss is removed, the MFSN-FRSE's performance will also decrease. Introducing the difference loss can encourage the model to separate the private features from the common feature, leading to better DA performance. However, in RI → WA1 and RI → AB, if the difference loss is removed, the performance of MFSN-FRSE will increase. The main reason is that these two tasks have too few common matching features between the source and target domains. That is not enough to make a correct matching decision. We will try to tackle this issue in our future work.

In most tasks, if the similarity loss is removed, the model's performance will decrease significantly. Introducing the similarity loss can ensure the distributions of the source and target common matching features are similar. Therefore, the entity matching classifier can make correct decisions in both the source and target domains. We can also see that on the WDC dataset, introducing the similarity loss doesn't bring a great improvement. The possible reason is that the data distribution between the different WDC datasets is very similar [10].

To explore the optimal strategies for key modules in MFSN and MFSN-FRSE, we conduct a series of detailed tests in the appendix. The results show that the optimal similarity loss is the MMD loss, and the optimal pooling layer is the DomAttion mechanism. For the reconstruction and computing difference loss, the optimal strategy is to use the hidden representation of all tokens.

## 5 Conclusion

We propose a framework for DA in EM called **M**atching **F**eature **S**eparation **N**etwork (MFSN). Briefly, MFSN achieves good DA performance by explicitly modeling domain-specific features. It utilizes three Pre-LMs based encoders to learn the private and common matching features of the source and target domains. The difference loss can make the common and private matching features mutually orthogonal. The similarity loss can make the distributions of the both common matching features are similar. We also propose an enhanced variant called **F**eature **R**epresentation and **S**eparation **E**nhanced MFSN (MFSN-FRSE). Compared with MFSN, it has better feature representation and separation capabilities. It utilizes three enhanced encoders to learn more expressive private and common hidden representation matrices of both domains. Then, the difference loss is computed in a "token-by-token" manner, and the similarity loss is computed with the help of DomAtt. The experiment results show that our framework outperforms the previous SOTA methods. we verify the effectiveness of each module by ablation study. Finally, we explore the optimal strategy of each module by detailed tests.

**ACKNOWLEDGMENTS**

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

# A  Appendix

To explore the optimal strategies for key modules in MFSN and MFSN-FRSE, we conduct a series of detailed tests.

## A.1  Detailed test of similarity loss

Figure 6 and Figure 7 show the results of MFSN-basic and MFSN-FRSE using MMD loss, CoRAL loss, GRL loss, and without similarity loss. We can observe that the GRL-loss and MMD-loss have a strong generality, while the CORAL loss lacks flexibility. This is because it measures the discrepancy between two distributions only by their difference in the second-order statistics (a.k.a., the covariance). Therefore, it is only suitable for few datasets. For example, CORAL loss can effectively measure the discrepancy between Fodors-Zagats (FZ) and Zomato-Yelp (DZY). However, it can't accurately compute the discrepancy between Books2 (B2) and Fodors-Zagats (FZ). The adversarial training-based methods (e.g., GRL) learn a function that can reasonably compute the distribution differences according to given data examples, so it has a strong generality [24]. Equation (8) shows that MMD usually predefines multiple kernel functions $\varphi(\cdot)$ to measure the discrepancy between the source and target domains. In other words, MMD can measure the discrepancy from multiple perspectives, so it also has a strong generality.

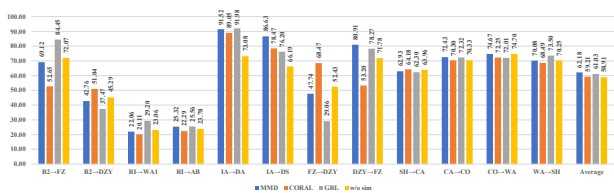

**Figure 6: The detailed test of similarity loss in MFSN-basic. W/o sim represents removing the similarity loss.**

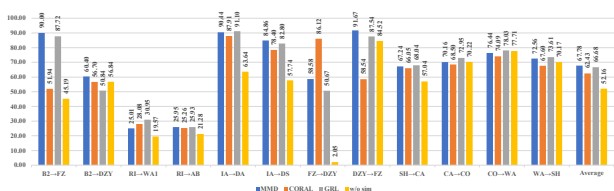

**Figure 7: The detailed test of similarity loss in MFSN-FRSE. W/o sim represents removing the similarity loss.**

## A.2  Detailed test of pooling strategy in MFSN-FRSE

To explore the optimal choice of pooling layer in MFSN-FRSE, we test the DomAtt and three different conventional pooling strategies. The results are shown in Figure 8. SelfAtt indicates using self-Attention for pooling: the hidden representation of [CLS] is the query, and the hidden representations of all tokens are the key and the value. CLS indicates taking the [CLS]'s hidden representation as the result of pooling. Mean indicates taking the average of all tokens' hidden representations as the result of pooling. From the overall results, DomAtt usually achieves better results. Compared with other methods, it can learn critical domain features from the input based on a domain-shared vector $a$. Therefore, DomAtt is more suitable for computing the similarity loss between two token sequences.

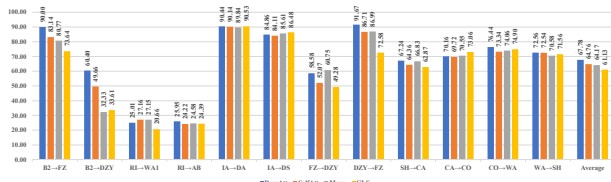

**Figure 8: The detailed test for pooling strategy. DomAtt, SelfAtt, Mean, and CLS indicate different pooling strategies.**

## A.3  Detailed test of Decoder

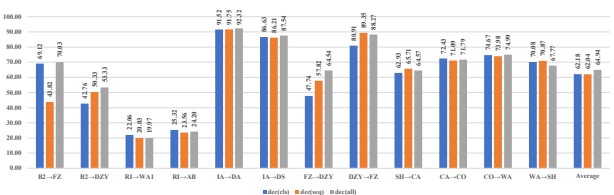

**Figure 9: The detailed test for the decoder in MFSN-basic. Dec(cls), dec(seq), and dec(all) represent the input strategies of the decoder. They correspond to only the [CLS] token, all tokens except for the [CLS] token, and all tokens in the sequence, respectively.**

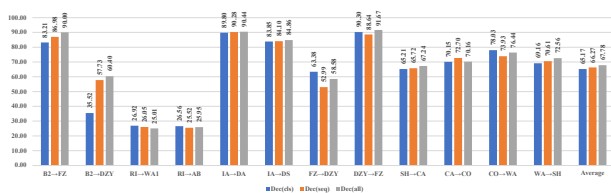

**Figure 10: The detailed test for the decoder in MFSN-FRSE. Dec(cls), dec(seq), and dec(all) represent the input strategies of the decoder. They correspond to only the [CLS] token, all tokens except for the [CLS] token, and all tokens in the sequence, respectively.**

To explore the optimal reconstruction strategy, we test the decoder to adopt three different strategies for reconstruction: only using the hidden representation of [CLS] (denoted as dec(cls)), using the hidden representation of all tokens except for [CLS] (denoted as dec(seq)), and using the hidden representation of all tokens (denoted as dec(all)). The results are shown in Figure 9 and Figure 10. Overall, the optimal reconstruction strategy for the decoder is dec(all). The results also show that dec(all) is better than dec(seq). This indicates that

[CLS] contains the semantic information of the whole record pair sequence, which is helpful for the reconstruction task. However, the effect of dec(cls) is often the worst, which indicates that the semantic information in [CLS] is relatively limited. Relying on [CLS] alone is not enough to reconstruct the whole sequence.

## A.4 Detailed test of Difference Loss

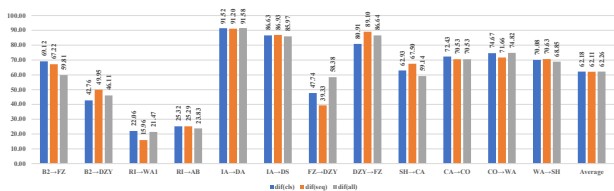

**Figure 11: The detailed test for the difference loss in MFSN-basic. dif(cls), dif(seq), and dif(all) represent three different strategies to calculate the difference loss. They correspond to only the [CLS] token, all tokens except for the [CLS] token, and all tokens in the sequence, respectively.**

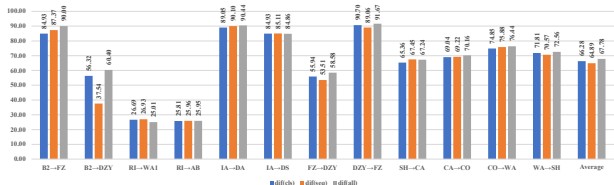

**Figure 12: The detailed test for the difference loss in MFSN-FRSE. dif(cls), dif(seq), and dif(all) represent three different strategies to calculate the difference loss. They correspond to only the [CLS] token, all tokens except for the [CLS] token, and all tokens in the sequence, respectively.**

Finally, we test three different strategies for computing the difference loss: only using the hidden representation of [CLS] (denoted as dif(cls)), using the hidden representations of all tokens except for [CLS] (denoted as dif(seq)), and using the hidden representations of all tokens (denoted as dif(all)). The results are shown in Figure 11 and Figure 12. We can observe that dif(all) is usually the optimal strategy, followed by dif(cls). This also indicates that the hidden representation of [CLS] can encoder the overall features of the record pairs sequence. Therefore, the model can achieve feature separation by computing the difference loss between the hidden representations of [CLS]. On this basis, introducing all the tokens in the sequence can further improve the feature separation ability of the model.