# OpenReview forum: "Matching Feature Separation Network for Domain Adaptation in Entity Matching"
_ACM.org/TheWebConf/2024/Conference — TheWebConf24_

### Official Review · Reviewer_gpPv · 2023-11-22

**Novelty:** 4
**Technical Quality:** 5

**Review:**

This paper looks at the problem of domain adaptation for entity matching. In particular, it argues for the use of language models to support domain adaptation in the process and proposes a new model architecture that tries to distinguish between features that are common across domains and those that are particular to the domains.  The shows good performance to published baselines but I was curious about performance against straight fine-tuned language models or even very large generative language models used in a few shot way. This has shown to have good performance on these entity matching tasks [1].


## Strong points

* S1) Domain adaption is an important tasks
* S2) The experimental setup for showing transfer performance is well done
* S3) The ablation study tests the contributions of each component architecture

## Weak points

* W1) While entity matching is interesting to web applications, the paper is not well situated with respect to the web.
* W2) The paper uses "older" language models (e.g. Bert) and does not motivate why to use these models
* W3) It was unclear the additional training data used for the adapted models. This was one of the arguments used to motivated the need for domain adaptation.

[1] Vos, David, Till Döhmen, and Sebastian Schelter. "Towards Parameter-Efficient Automation of Data Wrangling Tasks with Prefix-Tuning." NeurIPS 2022 First Table Representation Workshop. 2022.

#After rebuttal - I appreciate the authors response but my view is still the same.

**Questions:**

Can you comment on the amount of extra training data used during adaptation or are the models just trained and tested directly? It was unclear in the paper.

**Ethics Review Description:**

There are no ethics issues with this paper

**Reviewer Confidence:**

3: The reviewer is confident but not certain that the evaluation is correct

**Scope:**

2: The connection to the Web is incidental, e.g., use of Web data or API

---

### Official Review · Reviewer_Vjy9 · 2023-11-23

**Novelty:** 5
**Technical Quality:** 6

**Review:**

In this paper, the authors present an approach to domain adaptation in entity matching, addressing the challenges posed by data scarcity and the requirement for labeled data in specific domains. The central concept revolves around explicitly distinguishing common features shared by two domains from the domain-specific features, thereby discarding the latter to mitigate model confusion. The paper introduces two primary architectures. The first, MFSN, comprises encoder and decoder modules based on BERT to simultaneously learn both private and common features.

The second architecture, MFSN-FRSE, is positioned as an improved version of MFSN, incorporating hidden states of tokens that constitute an entity (including arguments and their labels).

According to the reported results, the proposed method demonstrates a noteworthy average improvement of 7% in F1 score compared to previous state-of-the-art approaches.

Pros:
* The inclusion of comprehensive technical details about the used architectures (MFSN/MFSN-FRSE) enhances the clarity and understanding of the proposed methodologies for training.
* The clarity of the training phase  allows readers to gain a deeper insight into the architecture, contributing to the transparency and reproducibility of the research.

Cons:
* Reference for Claims in Section 3.3: The assertion that the difference loss improves the insights of both shared and private encoders lacks proper reference support.

* Inference for Target Dataset Entity Matching: the paper lacks an adequate explanation of how the proposed model is applied during the inference phase for entity matching on a target dataset. To address this gap, the authors should include a detailed section or subsection explaining the steps involved in applying the model to new datasets for entity matching. This would provide readers with a more comprehensive understanding of the practical aspects of the proposed technique.

* Lack of Reference for DomAtt in Section 3.6: lacks proper references for DomAtt, creating ambiguity regarding the background and context of the discussed content.

Moreover, sharing the code would improve the replicability of this work

**Questions:**

* The training process is elucidated well; nevertheless, the methodology for obtaining results on the target domain database remains unclear. How is the inference carried out? Was there a section of the architecture dedicated to learning source domain matching features that was omitted?
* If the aim is to differentiate between common shared matching features and private matching features, the purpose of the difference loss becomes a point of inquiry. Is the intention to maximize similarity between shared features and common features within the same domain? If so, this poses a potential contradiction, as the primary goal is to distinguish these features rather than emphasize their similarity.

**Ethics Review Description:**

n.a.

**Reviewer Confidence:**

3: The reviewer is confident but not certain that the evaluation is correct

**Scope:**

4: The work is relevant to the Web and to the track, and is of broad interest to the community

---

### Official Review · Reviewer_fDPu · 2023-11-24

**Novelty:** 5
**Technical Quality:** 6

**Review:**

The paper describes an approach to the Entity Matching problem, i.e. the problem of identifying when two different records in different data sets describe the same entity in the real world.

Machine learning techniques for entity matching are well explored, but suffer from the necessity of labelling prohibitive amounts of training data.  To address this, an elaborate framework for this task is presented, that attempts to use transfer learning techniques to achieve good performance on domain specific data sets by leveraging pre-trained general purpose language models.

The framework is presented in detail and an experimental evaluation is given.

Although entity matching is a problem for some applications of semantic technologies, I find this paper to match the topics of the Semantics and Knowledge track only marginally.

**Questions:**

The framework seems to rely heavily on pre-trained *language* models.  Is it implicitly assumed that the datasets in question consist (largely) of natural language?

Or would the approach work for traditional relational databases?

How about knowledge graphs?

**Ethics Review Description:**

no issue

**Reviewer Confidence:**

2: The reviewer is willing to defend the evaluation, but it is likely that the reviewer did not understand parts of the paper

**Scope:**

2: The connection to the Web is incidental, e.g., use of Web data or API

---

### Official Review · Reviewer_ePsi · 2023-11-25

**Novelty:** 2
**Technical Quality:** 4

**Review:**

This paper proposes a new domain adaptation (DA) framework named MFSN for entity matching (EM). The framework employs three encoders to obtain matching features. One shared encoder is used to obtain common features between the source domain and the target domain; two individual encoders are used to obtain private features for the two domains, respectively. The authors design a similarity loss and a difference loss to distinguish the common and private features. Furthermore, a decoder is introduced to reconstruct candidate pairs based on their features, which ensures the features are relevant to the EM task. Finally, candidate pairs are classified by an MLP. Massive experiments are conducted to demonstrate the effectiveness of the proposed framework, including 12 transferring EM tasks, the visualization analysis, and the ablation study.

Strengths:

S1. This paper focuses on an important issue: how to perform entity matching with unlabeled data, which is overlooked by existing supervised works.

S2. The authors conduct expensive experiments on 12 tasks, and the results are sufficient to demonstrate the effectiveness and generalization of the proposed method.

S3. This paper is well-written and has good readability.

Weaknesses:

W1. The method lacks novelty. The framework for separating private and common matching features is proposed in the related work [19]. The authors just replace the encoders with pre-trained BERT, and replace the decoder with Transformer-Decoder. Figure 2 in this paper is very similar to Fig. 2 in [19].

W2. The analyses are insufficient. Besides the main results, only visualization and ablation studies are provided.

**Questions:**

Q1. Typos: In the last sentence of Introduction, Section 6 => Section 5

Q2. I recommend specifying that the classifier in Equation (17) is an MLP, although this can be taken from Figure 2.

Q3. In Equation (14), how to compute the cross-entropy loss between two tokens?

Q4. How to determine the best hyper-parameters? There are three hyper-parameters in Equation (25). In my opinion, it is very hard to find the best one.

Q5. How is visualization performed? For each candidate pair, MFSN-FRSE obtains a common feature and a private feature, which one is selected for visualization, or they are summed up?

Q6. It can be seen that massive experiments are conducted in the appendix. However, the analyses are insufficient in Section 4. For example, this paper emphasizes separating common features and private features, will the private feature distributions of the source domain and the target domain be obviously different while the distributions of common features are similar? Would you mind providing some analyses about this?

**Reviewer Confidence:**

3: The reviewer is confident but not certain that the evaluation is correct

**Scope:**

3: The work is somewhat relevant to the Web and to the track, and is of narrow interest to a sub-community

---

### Decision · Program_Chairs · 2024-01-22

**Decision:**

Accept

**Comment:**

This paper addresses the important task (for KG construction) of entity matching when there is data scarcity, specifically exploring domain adaptation using pre-trained language models like BERT. The advantages lie in its focus on entity matching with unlabeled data, extensive experiments on 12 tasks to demonstrate the effectiveness (compared to previous state-of-the-art approaches) and generalization , and a well-structured presentation. The experimental setup and ablation studies contribute to the paper's strength.

 The authors addressed some of the concerns on weaknesses and questions from the reviewers, including a perceived lack of novelty with the method building on an existing framework and, while the system shows good performance against baselines, it is not compared to more recent LLMs. The use of older language models like BERT raised concerns about the performance against straight fine-tuned language models or even very large generative language models. Nonetheless, authors see the pre-trained LM as just one component to implement of the encoder with the research goal is to explore how to effectively combine pre-trained LMs with domain separation networks to solve the domain adaptation in entity matching

 In the camera-ready version, authors should include the answers to reviewers regarding addressing the novelty concerns, providing clearer analyses, and expanding the discussion and clarifications, as well as providing the source code.